# Sleep Quality as a Mediator of Burnout, Stress and Multisite Musculoskeletal Pain in Healthcare Workers: A Longitudinal Study

**DOI:** 10.3390/healthcare11182476

**Published:** 2023-09-06

**Authors:** Ludmilla Maria Souza Mattos de Araújo Vieira, Vivian Aline Mininel, Tatiana de Oliveira Sato

**Affiliations:** 1Physical Therapy Department, Universidade Federal de São Carlos, São Carlos 13565-905, Brazil; ludmilla@estudante.ufscar.br; 2Nursing Department, Universidade Federal de São Carlos, São Carlos 13565-905, Brazil; vivian.aline@ufscar.br

**Keywords:** mediation analysis, cohort study, multisite pain

## Abstract

This study aimed to verify whether sleep quality is a mediator of the relationship between burnout, stress and multisite pain in healthcare workers in a longitudinal study with 12 months of follow-up during the COVID-19 pandemic. Online questionnaires were used for data collection. The sociodemographic questionnaire contained personal and occupational data. The short version of the Copenhagen Psychosocial Questionnaire (COPSOQ II-Br) was used to assess burnout and stress. The Pittsburgh Sleep Quality Index (PSQI-Br) assessed sleep quality over one month. The Nordic Musculoskeletal Questionnaire (NMQ) aimed to identify the presence of multisite musculoskeletal pain in the last 12 months and the last 7 days. Mediation analysis was used to verify whether the effect of the predictor variables (burnout and stress) on the dependent variable (number of pain sites in the last 12 months and 7 days) was due to an intervening variable (sleep quality). Stress and burnout were associated with multisite pain in healthcare workers in the last 12 months and the last 7 days. Sleep quality was a significant mediator of this association in the last 12 months, indicating that some of the association between burnout, stress and multisite pain can be explained by poor sleep quality. Thus, a comprehensive approach to long-term multisite pain should consider psychosocial aspects such as burnout, stress and sleep quality.

## 1. Introduction

The health status of healthcare workers can affect their quality of life and the treatment offered to their patients. The COVID-19 pandemic has brought this issue to the forefront, given the high work demands and rapid changes in this sector. Physical, mental and emotional overload of healthcare workers has always existed [1]; however, events such as outbreaks and pandemics deserve attention [2].

Burnout symptoms can be an expression of emotional overload, where the response to work-related chronic stress causes damage at cognitive and emotional levels due to extreme exhaustion, stress and physical exhaustion resulting from work situations [3]. Burnout symptoms can differ within and between individuals [4] and can lead to sick leave and absence from work [5]. A systematic review of medical students in Brazil before the pandemic identified the prevalence of burnout (13%), stress (50%), poor sleep quality (52%) and excessive daytime sleepiness (46%) [6]. Another study of physicians conducted before the pandemic found that anesthesiologists affected by burnout report a loss of efficiency and less value for their achievements, but still consider that they do a good job compared with non-medical professionals [7]. Systematic reviews of healthcare workers during global infectious disease outbreaks, for example, COVID-19, SARS, H1N1, EBOLA and MERS found high incidences of burnout (28–32%), acute stress (30%), post-traumatic stress disorder (13%) and insomnia (40%) [2,6,8].

Another condition that can cause emotional and physical exhaustion is occupational stress. Theories about stress emphasize the influence of psychic factors on bodily functions [9], leading to a negative or positive response in the stressor, which may have an impact on mental or physical health due to the balance between perceived demands and individuals’ resources and skills [10]. A study in Argentina (2021) found a prevalence of stress (94%) and burnout (74%) in physicians [11]. In the USA, another study classified healthcare workers into those with high or very high daily stress (30%) and those with high or very high burnout (49%) [12]. Other studies also found high rates of burnout (57%, 49% and 44%), emotional exhaustion (36%, 29% and 92%), and depersonalization (58%, 18% and 27%) in this population [13,14,15].

Relationships between burnout, stress and musculoskeletal symptoms are present in the literature [16,17]. In healthcare workers, general stress and work-related stress were considered strong risk factors for musculoskeletal disorders [17].

More than half of nurses in a study conducted in Vietnam had musculoskeletal symptoms in the last 12 months, with a prevalence of 51% in men and 58% in women [18]. Physicians and other healthcare workers also had high incidences of musculoskeletal pain in the neck (60%), shoulders (52%) and lower back (49%) [19,20,21].

In addition to burnout being considered an important predictor of musculoskeletal pain [22], it can also interfere with sleep quality [5]. Sleep disturbances are important causative and maintaining factors for burnout [23] and sleep physiology improves following recovery from burnout [24]. 

Stress can also interfere with sleep quality. A study of healthcare workers conducted in India during the pandemic found that 36% of workers had poor sleep quality, while 25% had moderate burnout [25]. Poor sleep and work-related stress can interfere with patient care [26]. Nurses with high levels of stress also report lower sleep quality [27], while nurses with high levels of burnout and traumatic stress are more likely to have physical and mental health problems, with sleep disturbance being a partial mediator of these relationships [28]. 

Insomnia has also been associated with pain and sleep disturbances in various body parts in nurses [29,30]. Low back pain is associated with lower sleep quality [31]. Nurses who sleep for less than 7 h have more chronic discomfort in the neck and shoulders [32] and patients with chronic musculoskeletal pain report more insomnia [33]. 

Mediation analysis can clarify the relationship between variables and outcomes, as the measurement model is important for understanding these mechanisms. The mediating variable that accounts for the relationship between a predictor and an outcome can be a qualitative or quantitative one and can fully explain the relationship (full mediation) or explain the relationship to a limited degree only (partial mediation), assuming a three-variable system (Figure 1). In this model, two existing paths can lead to the outcome (dependent) variable: the direct effect of the independent variable on the dependent variable and the effect of the mediating variable on the dependent variable, and a path that connects the independent variable to the mediating variable [34,35,36].

However, the mediation relationship between psychosocial aspects and musculoskeletal pain is inconclusive [37]. Considering that we did not find similar studies in the literature, it is important to examine relationships between burnout, stress, sleep quality and multisite musculoskeletal pain in the last 12 months and the last 7 days. Therefore, the objective of this study was to verify whether sleep quality is a full or partial mediator of the relationship between burnout, stress and multisite pain in healthcare workers in a longitudinal study with a 12-month follow-up during the COVID-19 pandemic.

## 2. Materials and Methods

### 2.1. Study Design

This was a longitudinal study of healthcare workers in Brazil conducted during the COVID-19 pandemic between 19 June 2021 and 18 December 2022. The STrengthening the Reporting of OBservational Studies in Epidemiology (STROBE) statement checklist was followed [38].

### 2.2. Participants

We analyzed data from the HEROES (HEalth conditions of healthcaRe wOrkErS—HEROES) cohort composed of 125 healthcare workers [39]. For this study, baseline and quarterly follow-up data collected over 12 months were considered. The project was approved by the Research Ethics Committee (CAAE: 39705320.9.0000.5504).

Recruitment of participants was conducted through the Internet, using social networks and e-mails available on institutional websites. Posts about the project were placed on Instagram, Facebook and YouTube. In addition, e-mails were sent to public hospitals, Departments and Health Units for dissemination.

The inclusion criteria were: healthcare worker in the National Health System, aged between 18 and 60 years old and active in care activities. Participation was voluntary and there was no financial incentive. Students, retirees, duplicate responses and inconsistent data were excluded.

### 2.3. Data Collection Instruments

Online questionnaires were used for data collection. The sociodemographic questionnaire contained personal and occupational data, including age, sex, marital status, education, smoking, use of medication and working hours.

The short version of the Copenhagen Psychosocial Questionnaire (COPSOQ II-Br) was used to assess burnout and stress [40,41,42]. This questionnaire consists of 23 dimensions and 40 questions [40]. The questions are related to the last four weeks and are scored based on a Likert scale with 5 response options. The dimensions of burnout (18A + 18B) and stress (19A + 19B) are scored as follows: 0: not at all; 1: a small part of the time; 2: part of the time; 3: a large part of the time; 4: all the time. The scores range from 0 to 8. Burnout and stress were calculated by summing the answers to the individual questions for each dimension [40].

The Pittsburgh Sleep Quality Index (PSQI-Br) assesses sleep quality and disturbances over one month. It consists of 19 questions whose responses are obtained through self-reporting and 5 questions that are directed at the spouse or roommate. The final 5 questions are used only for clinical practice and do not contribute to the total index score. The 19 questions are categorized into 7 components, rated on scores from 0 (no difficulty) to 3 (severe difficulty). The PSQI components are subjective sleep quality (C1), sleep latency (C2), sleep duration (C3), habitual sleep efficiency (C4), sleep disturbances (C5), use of sleeping medication (C6) and daytime sleep dysfunction (C7). The sums of the values attributed to the 7 components range from 0 to 21 and the higher the number, the worse the sleep quality [43]. The PSQI-Br was translated and validated in the Brazilian population [44].

The Nordic Musculoskeletal Questionnaire (NMQ) aims to identify the presence of musculoskeletal symptoms in different body regions: neck, shoulders, elbows, wrists/hands, thoracic spine, lumbar spine, hip, knees and ankles/feet [45]. It assesses the presence of symptoms, the occurrence of functional disability and seeking help from a health professional 12 months and/or 7 days prior to data collection. The answers have a dichotomous characteristic (presence or absence). For this study, we used the validated NMQ Brazilian version [46] and analyzed only data on symptoms in the last 12 months and the last 7 days.

### 2.4. Data Analysis

Data were analyzed descriptively (frequencies, means, standard deviation and confidence intervals) using SPSS software (version 23.0). Mediation analysis was performed using the Process macro developed by Hayes [34] in the SPSS statistical program. The dependent variable (Y) was the number of regions with pain in the last 12 months and the last 7 days, ranging from 0 to 9 sites. The independent or predictive variables (X) were burnout and stress (ranging from 0 to 8 points) included in separate models (Figure 1). The mediating variable (M) was the total sleep quality score (ranging from 0 to 21 points).

Mediation analysis enables the assessment of whether the effect of a predictor variable (burnout and stress) on a dependent variable (number of regions with pain) is due to an intervening variable (sleep quality). The results of the mediation analysis are reported using the total effect (c), indirect effect (a*b) and direct effect (c’). The indirect effect (a*b) is the effect of the predictor variable X on the dependent variable Y mediated by M. The direct effect (c’) is the effect of X on Y controlled by the mediator M. The total effect is the sum of direct and indirect effects. The mediated effect, that is, the relative contribution of the mediator, was also calculated.

The assumptions of the mediation analysis were tested, namely: X must predict Y, X must predict M, M must predict Y and X should predict weaker Y after adding M [34]. Bootstrapping is a nonparametric approach to statistical inference that produces standard errors and confidence intervals (CIs) for each of the parameters. Bootstrapping was performed using 5000 resampling steps and 95% CIs for bias correction to investigate indirect effects [34].

The sample size was calculated a posteriori, based on the formula proposed by Fritz and MacKinnon [47]. The formula was applied considering a type I error of 5%, a power of 80%, and one predictor. The sample size varied based on the coefficient obtained in each part of the mediation model, varying from 7 to 78,500 participants. The relationship between sleep and pain was underpowered for all tested associations, with the minimum required number of participants being 274. 

## 3. Results

The healthcare workers (n = 125) who participated in the study aged 38 years were female (83%), married (57%), had completed higher education (80%), used medication (66%), composed the nursing team (58%), worked in the hospital (49%) and had a weekly workload of more than 30 h (71%). The mean values for burnout and stress were high (Table 1). Comparison between groups with no pain/single-site pain and multisite pain identified differences in burnout and stress, with a higher mean in the multisite pain group. 

The components of the Pittsburg Sleep Quality Index are presented in Table 2. Most workers reported good subjective sleep quality (58%), acceptable sleep latency (52%), 6 or more hours of sleep (54%), >85% sleep efficiency (62%), up to two sleep disturbances (90%), no use of sleeping medication in the last month (66%) and one daytime dysfunction (58%). Most healthcare workers were classified as poor sleepers (74%). Comparison between groups with no pain/single-site pain and multisite pain identified differences in all components of the PSQI, except sleep duration and habitual sleep efficiency, with worse indices in the multisite pain group.

The number of pain sites in the last 12 months and the last 7 days are presented in Table 3. Most workers reported between 4 and 6 pain sites in the last 12 months and between 0 and 2 pain sites in the last 7 days. 

The results of the mediation analysis are presented in Table 4 and Table 5. The associations between burnout and multisite pain (c) and between burnout and sleep quality (a) were significant (*p* < 0.01) at all time points evaluated in the last 12 months. Sleep quality was significantly associated with multisite pain (b) at baseline only (*p* = 0.04). The association between burnout and pain was significant and weaker after the addition of the mediator (c’) at all the time points evaluated. The mediated effect ranged from 13.6 to 20.4%, indicating a partial mediation effect. For the last 7 days, the associations between burnout and multisite pain (c) and between burnout and sleep quality (a) were significant at all evaluated time points (*p* < 0.01). However, sleep quality was not significantly associated with multisite pain (b) (*p* > 0.05). For all time points evaluated, the association between burnout and pain (c’) was weaker after the addition of the mediator, although the mediation effect was not significant.

The associations between stress and 12-month multisite pain (c) and between stress and sleep (a) were significant at all time points evaluated (*p* < 0.01). Sleep quality was significantly associated with 12-month multisite pain (b) at all assessment times, except at the 9-month follow-up (*p* = 0.07). The association between stress and pain (c’) was significant and weaker after the addition of the mediator at all the time points evaluated. The mediated effect ranged from 21.8 to 30.6%, indicating a partial mediation effect.

For the last 7 days, the associations between stress and multisite pain (c) and between stress and sleep (a) were significant at all time points evaluated (*p* < 0.01), except at 3 months. Sleep quality was not significantly associated with multisite pain (b) at all assessment times (*p* > 0.05). The association between stress and pain (c’) was weaker after the addition of the mediator at all the time points evaluated, and the mediation effect was significant at baseline, 6 months and 12 months (Table 5).

## 4. Discussion

Stress and burnout were significantly associated with multisite pain in the last 12 months and the last 7 days in healthcare workers. Burnout and stress were also significantly associated with sleep quality. Sleep quality was a significant mediator of the relationship between stress and multisite pain in the last 12 months. However, no mediation effect was observed when pain in the last 7 days was analyzed.

Healthcare workers suffer from burnout and stress, with high mean levels of burnout (5.3) and stress (4.9) recorded. Burnout and stress are considered predictors of musculoskeletal pain [22,48] and poor sleep quality [49], but the mediating effect of sleep quality in multisite pain was not clear. This finding can be explained by methodological aspects such as the small sample size used in the follow-up measurements.

A systematic meta-analysis [49] found a positive and significant correlation between high levels of burnout and poor quality of sleep in nurses. The dissatisfaction with sleep quality was also associated with burnout in shift workers [50].

Most healthcare workers were classified as bad sleepers (74%); this was similar to results obtained in Colombian healthcare workers [51]. Although healthcare workers may be more propense to sleep problems due to the nature of their work, a systematic review of Western healthcare workers showed high levels of stress, sleep disturbances, burnout and other illnesses, with more frequent and intense symptoms in women and frontline nurses working during the COVID-19 pandemic [52].

Stress at work has a negative impact on sleep quality [53]. A two-year follow-up study of healthcare workers also showed that stress levels increased and sleep quality worsened during the pandemic [54]. The systematic review by Dragioti [55] corroborates these findings, as healthcare workers affected by COVID-19 suffered from stress and sleep problems (33% and 37%).

Sleep disturbances are associated with pain [56] and sleep deficiencies predict incidences and exacerbations of chronic pain [57], where deprivation and sleep interruptions increase sensitivity and vulnerability to pain and can form a vicious cycle where they maintain and increase each other [58]. Restricting sleep to 6 h can increase pro-inflammatory cytokines [59]. Sleep deprivation is also associated with cognitive impairment [60]. A systematic review of military personnel revealed the association between insufficient sleep and musculoskeletal injuries [61].

A cohort study also showed that stress, non-restorative sleep and physical inactivity are risk factors for the development of chronic pain [62]. Using mediation analysis, Walton et al. [63] found that the association between perceived stress and pain intensity was wholly mediated by sleep interference [63]. Another study investigating the environmental factors that affect the well-being of critical care surgeons found that they were at increased risk of sleep deprivation, musculoskeletal pain and injuries and burnout [64], where sleep-associated problems seemed to precede low back pain and exhaustion in the working population [65]. Thus, it seems that stress and sleep management, as well as ergonomic interventions, post the COVID-19 pandemic are urgent in healthcare services and can help to improve musculoskeletal health.

### Limitations and Strengths of the Study

This study provides information on the role of sleep quality as a mediator of the association between burnout, stress and multisite pain in health professionals; however, some limitations must be considered when interpreting the findings. Although the longitudinal nature of the study is a strength, the small sample size and the lack of a priori sample size calculation are limitations. The use of questionnaires to obtain data on burnout, stress and sleep quality can also be a limitation. Specific questionnaires are suggested to assess stress and burnout.

## 5. Conclusions

Burnout, stress, sleep quality and multisite musculoskeletal pain are topics relevant to healthcare workers in the COVID-19 pandemic context.

Our findings showed that sleep quality mediates the association between stress, burnout and multisite pain in Brazilian healthcare workers in the previous 12 months. Thus, this study highlights the need for a comprehensive approach, which also considers psychosocial work aspects and sleep quality, to multisite pain management.

## Figures and Tables

**Figure 1 healthcare-11-02476-f001:**
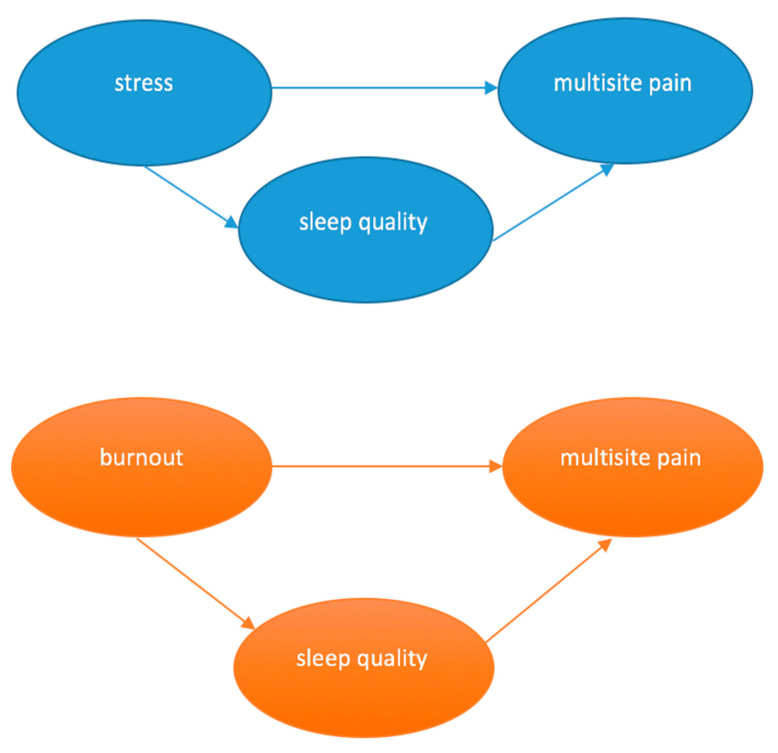
Schematic representation of stress and burnout as multisite pain predictors mediated by sleep quality based on Hayes’s measurement model 4 [34].

**Table 1 healthcare-11-02476-t001:** Sociodemographic and personal data of the healthcare workers at baseline (n = 125).

Characteristics	Total	No Pain/Single-Site Pain (n = 24)	Multisite Pain (n = 101)	*p*
	n (%)	n (%)	n (%)	
Age, years ^1^	37.5 (8.3)	39.6 (9.0)	36.9 (8.1)	0.16
Sex				0.07
female	104 (83.2)	17 (70.8)	87 (86.1)	
male	21 (16.8)	7 (29.2)	14 (13.9)	
Marital status				0.87
single/widower/divorced	54 (43.2)	10 (41.7)	44 (43.6)	
married/with a partner	71 (56.8)	14 (58.3)	57 (56.4)	
Education level				0.50
Primary/high school education	25 (20.0)	6 (25.0)	19 (18.8)	
University	100 (80.0)	18 (75.0)	82 (81.2)	
Use medication	83 (66.4)	16 (66.7)	67 (66.3)	0.98
Smoke	14 (11.2)	3 (12.5)	11 (10.9)	0.82
Occupation				0.07
nurse	45 (36.0)	9 (37.5)	36 (35.6)	
nurse technician/auxiliary	28 (22.4)	4 (16.7)	24 (23.80	
physical therapist	26 (20.8)	2 (8.3)	24 (23.8)	
physician	8 (6.4)	3 (12.5)	5 (5.0)	
dentist	4 (3.2)	0 (0.0)	4 (4.0)	
other	14 (11.2)	6 (25.0)	8 (7.9)	
Workplace				0.39
primary care	40 (32.0)	7 (29.2)	33 (32.7)	
hospital	61 (48.8)	9 (37.5)	52 (51.5)	
emergency care	12 (9.6)	5 (20.8)	7 (6.9)	
ambulatorial care	4 (3.2)	1 (4.2)	3 (3.0)	
psychosocial care	5 (4.0)	1 (4.2)	4 (4.0)	
home care	3 (2.4)	1 (4.2)	2 (2.0)	
Weekly working hours				0.65
up to 30 h	36 (28.8)	6 (16.7)	18 (20.2)	
more than 30 h	89 (71.2)	30 (83.3)	71 (79.8)	
Burnout (0–8 points) ^1^	5.3 (1.8)	3.9 (1.3)	5.6 (1.8)	<0.01
Stress (0–8 points) ^1^	4.9 (1.9)	3.6 (1.3)	5.3 (1.8)	<0.01

^1^ Mean and standard deviation.

**Table 2 healthcare-11-02476-t002:** Components of the Pittsburg Sleep Quality Index (PSQI) at baseline (n = 125).

Pittsburg Sleep Quality Index (PSQI) Components	Total	No Pain/Single-Site Pain (n = 24)	Multisite Pain (n = 101)	*p*
	n (%)	n (%)	n (%)	
Subjective sleep quality				<0.01
very good	13 (10.4)	4 (16.7)	9 (8.9)	
fairly good	60 (48.0)	18 (75.0)	42 (41.6)	
fairly bad	38 (30.4)	2 (8.3)	36 (35.6)	
very bad	14 (11.2)	0 (0.0)	14 (13.9)	
Sleep latency				<0.01
0	16 (12.8)	5 (20.8)	11 (10.9)	
1	49 (39.2)	16 (66.7)	33 (32.7)	
2	29 (23.2)	2 (8.3)	27 (26.7)	
3	31 (24.8)	1 (4.2)	30 (29.7)	
Sleep duration				0.21
>7 h	33 (26.4)	10 (41.7)	23 (22.8)	
6–7 h	35 (28.0)	6 (25.0)	29 (28.7)	
5–6 h	33 (26.4)	6 (25.0)	27 (26.7)	
<5 h	24 (19.2)	2 (8.3)	22 (21.80	
Habitual sleep efficiency				0.11
>85%	77 (61.6)	20 (83.3)	57 (56.4)	
75–84%	24 (19.2)	2 (8.3)	22 (21.8)	
65–74%	10 (8.0)	1 (4.2)	9 (8.9)	
<65%	14 (11.2)	1 (4.2)	13 (12.9)	
Sleep disturbances				0.03
0	1 (0.8)	1 (4.2)	0 (0.0)	
1	55 (44.0)	15 (62.5)	40 (39.6)	
2	57 (45.6)	7 (29.2)	50 (49.5)	
3	12 (9.6)	1 (4.2)	11 (10.9)	
Use of sleeping medication				0.05
none	82 (65.6)	20 (83.3)	62 (61.4)	
less than once a week	17 (13.6)	4 (16.7)	13 (12.9)	
once or twice a week	9 (7.2)	0 (0.0)	9 (8.9)	
3 or more times a week	17 (13.6)	0 (0.0)	17 (16.8)	
Daytime dysfunction				0.02
0	17 (13.6)	7 (29.2)	10 (9.9)	
1	55 (44.0)	12 (50.0)	43 (42.6)	
2	42 (33.6)	3 (12.5)	39 (38.6)	
3	11 (8.8)	2 (8.3)	9 (8.9)	
PSQI score ^1^	8.80 (4.14)	5.67 (2.41)	9.54 (4.13)	<0.01
PSQI category				<0.01
good sleepers	32 (25.6)	12 (50.0)	20 (19.8)	
bad sleepers	93 (74.4)	12 (50.0)	81 (80.2)	

^1^ Mean and standard deviation.

**Table 3 healthcare-11-02476-t003:** Number of pain sites in the last 12 months and the last 7 days at baseline and follow-up.

Number of Pain Sites	Baseline n = 125	3 Months n = 80	6 Months n = 65	9 Months n = 63	12 Months n = 61
Last 12 months	n (%)	n (%)	n (%)	n (%)	n (%)
0	14 (11.2)	4 (5.0)	5 (7.7)	6 (9.5)	10 (16.4)
1	10 (8.0)	4 (5.0)	4 (6.2)	7 (11.1)	4 (6.6)
2	13 (10.4)	10 (12.5)	10 (15.4)	4 (6.3)	8 (13.1)
3	12 (9.6)	11 (13.8)	6 (9.2)	9 (14.3)	11 (18.0)
4	21 (16.8)	17 (21.3)	18 (27.7)	9 (14.3)	9 (14.8)
5	22 (17.6)	9 (11.3)	9 (13.8)	10 (15.9)	7 (11.5)
6	13 (10.4)	9 (11.3)	6 (9.2)	8 (12.7)	6 (9.8)
7	11 (8.8)	8 (10.0)	4 (6.2)	3 (4.8)	2 (3.3)
8	7 (5.6)	8 (10.0)	2 (3.1)	5 (7.9)	4 (6.6)
9	2 (1.6)	-	1 (1.5)	2 (3.2)	-
Last 7 days					
0	35 (28.0)	29 (36.3)	22 (33.8)	22 (34.9)	22 (36.1)
1	27 (21.6)	12 (15.0)	17 (26.2)	13 (20.6)	16 (26.2)
2	19 (15.2)	14 (17.5)	9 (13.8)	12 (19.0)	10 (16.4)
3	19 (15.2)	11 (13.8)	5 (7.7)	7 (11.1)	5 (8.2)
4	12 (9.6)	7 (8.8)	9 (13.8)	3 (4.8)	2 (3.3)
5	7 (5.6)	3 (3.8)	2 (3.1)	5 (7.9)	2 (3.3)
6	3 (2.4)	1 (1.3)	1 (1.5)	-	2 (3.3)
7	3 (2.4)	2 (2.5)	-	-	-
8	-	1 (1.3)	-	1 (1.6)	2 (3.3)
9	-	-	-	-	-

**Table 4 healthcare-11-02476-t004:** Mediation analysis between burnout and multisite pain in the last 12 months and the last 7 days, with sleep quality as the mediator.

Predictor: Burnout	Burnout–Pain (c)	Burnout–Sleep (a)	Sleep–Pain (b)	Burnout–Sleep–Pain (c’)
Pain in the last 12 months				
*baseline (n = 125)*				
mediated effect	20.4%			
b	0.65	1.19	0.11	0.52
95% CI	0.44–0.86	0.84–1.54	0.01–0.21	0.28–0.76
R^2^	24%	27%	26%	26%
*p*	<0.01	<0.01	0.04	<0.01
*3 months (n = 80)*				
mediated effect	14.8%			
b	0.65	1.32	0.07	0.56
95% CI	0.42–0.88	0.87–1.76	−0.05–0.19	0.28–0.84
R^2^	29%	31%	30%	30%
*p*	<0.01	<0.01	0.24	<0.01
*6 months (n = 65)*				
mediated effect	19.3%			
b	0.69	1.35	0.10	0.56
95% CI	0.44–0.94	0.82–1.88	−0.02–0.21	0.27–0.85
R^2^	33%	29%	36%	36%
*p*	<0.01	<0.01	0.10	<0.01
*9 months (n = 63)*				
mediated effect	13.6%			
b	0.75	1.19	0.08	0.66
95% CI	0.48–1.01	0.72–1.66	−0.07–0.22	0.34–0.97
R^2^	34%	29%	36%	36%
*p*	<0.01	<0.01	0.29	<0.01
*12 months (n = 61)*				
mediated effect	18.6%			
b	0.74	1.24	0.11	0.60
95% CI	0.48–1.00	0.75–1.73	−0.02–0.25	0.29–0.90
R^2^	35%	30%	38%	38%
*p*	<0.01	<0.01	0.10	<0.01
Pain in the last 7 days				
*baseline (n = 125)*				
b	0.48	1.19	0.02	0.45
95% CI	0.32–0.64	0.84–1.54	−0.06–0.11	0.26–0.64
R^2^	22%	27%	22%	22%
*p*	<0.01	<0.01	0.55	<0.01
*3 months (n = 80)*				
b	0.35	1.32	−0.01	0.35
95% CI	0.12–0.57	0.87–1.76	−0.12–0.11	0.08–0.63
R^2^	11%	31%	11%	11%
*p*	<0.01	<0.01	0.92	<0.01
*6 months (n = 65)*				
b	0.31	1.35	0.03	0.27
95% CI	0.09–0.53	0.82–1.88	−0.08–0.13	0.01–0.53
R^2^	11%	29%	11%	11%
*p*	0.01	<0.01	0.62	0.04
*9 months (n = 63)*				
b	0.29	1.19	0.07	0.21
95% CI	0.07–0.51	0.72–1.66	−0.05–0.19	−0.05–0.47
R^2^	10%	29%	12%	12%
*p*	0.01	<0.01	0.26	0.12
*12 months (n = 61)*				
b	0.40	1.24	0.03	0.36
95% CI	0.15–0.65	0.75–1.73	−0.11–0.16	0.06–0.66
R^2^	15%	30%	15%	15%
*p*	<0.01	<0.01	0.68	0.02

**Table 5 healthcare-11-02476-t005:** Mediation analysis between stress and multisite pain in the last 12 months and the last 7 days, with sleep quality as the mediator.

Predictor: Stress	Stress–Pain (c)	Stress–Sleep (a)	Sleep–Pain (b)	Stress–Sleep–Pain (c’)
Pain in the last 12 months				
*baseline (n = 125)*				
mediated effect	27.0%			
b	0.54	1.01	0.15	0.40
95% CI	0.33–0.75	0.66–1.37	0.04–0.25	0.17–0.63
R^2^	18%	20%	23%	23%
*p*	<0.01	<0.01	<0.01	<0.01
*3 months (n = 80)*				
mediated effect	25.0%			
b	0.56	1.23	0.12	0.41
95% CI	0.30–0.81	0.75–1.71	0.01–0.24	0.12–0.70
R^2^	19%	25%	23%	23%
*p*	<0.01	<0.01	0.05	0.01
*6 months (n = 65)*				
mediated effect	21.8%			
b	0.69	1.32	0.11	0.54
95% CI	0.42–0.95	0.76–1.89	0.00–0.23	0.24–0.84
R^2^	30%	26%	34%	34%
*p*	<0.01	<0.01	0.05	<0.01
*9 months (n = 63)*				
mediated effect	26.1%			
b	0.62	1.16	0.14	0.45
95% CI	0.31–0.92	0.65–1.67	−0.01–0.29	0.11–0.80
R^2^	21%	26%	25%	25%
*p*	<0.01	<0.01	0.07	0.01
*12 months (n = 61)*				
mediated effect	30.6%			
b	0.64	1.21	0.17	0.44
95% CI	0.34–0.94	0.67–1.74	0.03–0.31	0.11–0.77
R^2^	24%	26%	31%	31%
*p*	<0.01	<0.01	0.02	0.01
Pain in the last 7 days				
*baseline (n = 125)*				
b	0.39	1.01	0.06	0.33
95% CI	0.23–0.55	0.66–1.37	−0.02–0.14	0.15–0.51
R^2^	15%	20%	17%	17%
*p*	<0.01	<0.01	0.16	<0.01
*3 months (n = 80)*				
b	0.21	1.23	0.05	0.15
95% CI	−0.03–0.45	0.75–1.71	−0.07–0.16	−0.13–0.43
R^2^	4%	25%	5%	5%
*p*	0.09	<0.01	0.42	0.28
*6 months (n = 65)*				
b	0.34	1.32	0.02	0.31
95% CI	0.11–0.57	0.76–1.89	−0.08–0.13	0.04–0.57
R^2^	12%	26%	13%	13%
*p*	<0.01	<0.01	0.63	0.02
*9 months (n = 63)*				
b	0.24	1.16	0.09	0.14
95% CI	0.01–0.48	0.65–1.67	−0.03–0.21	−0.13–0.41
R^2^	7%	26%	10%	10%
*p*	0.04	<0.01	0.14	0.30
*12 months (n = 61)*				
b	0.43	1.21	0.04	0.38
95% CI	0.16–0.69	0.67–1.74	−0.09–0.16	0.08–0.69
R^2^	15%	26%	16%	16%
*p*	<0.01	<0.01	0.59	0.01

## Data Availability

The data that support the findings of this study are available on request from the corresponding author, T.d.O.S.

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
