# Peer review of "Sleep Quality as a Mediator of Burnout, Stress and Multisite Musculoskeletal Pain in Healthcare Workers: A Longitudinal Study"

_healthcare, 2023, doi:10.3390/healthcare11182476_

Round 1
Reviewer 1 Report
The authors have attempted to highlight the mediation role of sleep quality between burnout/stress and musculoskeletal pain. The presentation of the literature review should be presented in a more organised manner. Findings of multiple studies are merely listed, yet they should be compiled in a manner that flows where the aim of the study should be concise and the introduction should end with the motivation for performing this study. This is not apparent in the manuscript.
Baseline characteristics should be presented with mean values and SD. The 7 components of sleep analysis should be included in a table showing the number of participants selecting the different scores for all categories of sleep. The analysis of pain categories and sites should also be shown.
I do not think that the mediation of sleep quality has been made clear, does it have a direct or partial effect?
The manuscript flow and reading will be easier once grammar and editing of some of the sentences are improved.
Author Response
Reviewer #1
The authors have attempted to highlight the mediation role of sleep quality between burnout/stress and musculoskeletal pain. The presentation of the literature review should be presented in a more organised manner. Findings of multiple studies are merely listed, yet they should be compiled in a manner that flows where the aim of the study should be concise and the introduction should end with the motivation for performing this study. This is not apparent in the manuscript.
Author’s response: We appreciate the contributions and suggestions. We worked on the Introduction section to make it more fluid, clean and clear, facilitating understanding. Additional studies were included to improve our text (pages 1-2).
Baseline characteristics should be presented with mean values and SD. The 7 components of sleep analysis should be included in a table showing the number of participants selecting the different scores for all categories of sleep. The analysis of pain categories and sites should also be shown.
Author’s response: Thank you for your suggestion. We now included these data at Tables 2 and 3 (pages 5-7).
I do not think that the mediation of sleep quality has been made clear, does it have a direct or partial effect?
Author’s response: Thank you for your suggestion. The mediation analysis were reviewed and showed a partial effect. This information was added at Tables 4 and 5 (pages 7-10).
Comments on the Quality of English Language
The manuscript flow and reading will be easier once grammar and editing of some of the sentences are improved.
Author’s response: The manuscript was carefully reviewed to make the reading easier to the reader.
Reviewer 2 Report
Comment to the authors
Thank you for inviting me to review the manuscript entitled “Sleep quality as a mediator between burnout, stress, and multi-site musculoskeletal pain in healthcare workers – longitudinal study”.
This is a study that adopted a mediation analysis to investigate whether sleep quality is the mediator of stress and burnout with musculoskeletal pain. The topic is very important as it highlights the importance and the significance of sleep health for well-being, including pain and stress symptoms.
A shortcoming, that I am sure the authors have carefully thought about, is that the questionaries asssess different period of time. The PSQI investigates for sleep disturbances during the previous month. The NMQ assesses for the past 12 months and past 7 days, although from the method section, the authors seemed to have included only data from the previous 12 months. If that is the case, how can they run a mediation analysis investigating if poor sleep in the previous month can explain musculoskeletal pain symptoms in the past year? This is not clear and needs a good explanation. Maybe the recent musculoskeletal pain symptoms developed in the previous 7 days could be explained by poor sleep, as poor sleep has been suggested to occur prior to the development of pain. As for the assessment time period of the burnout and stress questionnaire, this has not been explicitly indicated.
Methods:
- as the period where the symptoms are reported is very important in this mediation analysis, it might be valuable to mention what does the Copenhagen Psychosocial Questionnaire refers to. For example, the authors correctly indicated that the questions of the PSQI refers to the previous month. Can the authors also include the time reference of the questions of the COPSOQ?
- it is not clear how the data at different time-point were obtained. Were the same subjects interviewed? Was the survey still been posted on the internet, social networks, etc?
Results:
- it would be clearer if the authors reiterated the total number of participants involved in the study in the first paragraph of the results.
Also, as part of the description of the sample size, it would be important to provide total value of PSQI, COPSOQ and NMQ.
Author Response
Thank you for inviting me to review the manuscript entitled “Sleep quality as a mediator between burnout, stress, and multi-site musculoskeletal pain in healthcare workers – longitudinal study”.
This is a study that adopted a mediation analysis to investigate whether sleep quality is the mediator of stress and burnout with musculoskeletal pain. The topic is very important as it highlights the importance and the significance of sleep health for well-being, including pain and stress symptoms.
Author’s response: Thank you for your comments. We agree with you about the relevance of this topic.
A shortcoming, that I am sure the authors have carefully thought about, is that the questionaries asssess different period of time. The PSQI investigates for sleep disturbances during the previous month. The NMQ assesses for the past 12 months and past 7 days, although from the method section, the authors seemed to have included only data from the previous 12 months. If that is the case, how can they run a mediation analysis investigating if poor sleep in the previous month can explain musculoskeletal pain symptoms in the past year? This is not clear and needs a good explanation. Maybe the recent musculoskeletal pain symptoms developed in the previous 7 days could be explained by poor sleep, as poor sleep has been suggested to occur prior to the development of pain. As for the assessment time period of the burnout and stress questionnaire, this has not been explicitly indicated.
Author’s response: Thank you very much for your suggestion. We now reviewed the analysis, including the 7 days symptoms (Tables 4 and 5).
Methods:
- as the period where the symptoms are reported is very important in this mediation analysis, it might be valuable to mention what does the Copenhagen Psychosocial Questionnaire refers to. For example, the authors correctly indicated that the questions of the PSQI refers to the previous month. Can the authors also include the time reference of the questions of the COPSOQ?
Author’s response: Thank you very much for your suggestion. We now included the time period for the COPSOQ evalaution at line 130 (page 3).
- it is not clear how the data at different time-point were obtained. Were the same subjects interviewed? Was the survey still been posted on the internet, social networks, etc?
Author’s response: Thank you for your comment. The same subjects were interviewed and data collection has finished yet (line 106).
Results:
- it would be clearer if the authors reiterated the total number of participants involved in the study in the first paragraph of the results.
Author’s response: We appreciate your suggestion. We included it at line 179 (page 4).
Also, as part of the description of the sample size, it would be important to provide total value of PSQI, COPSOQ and NMQ.
Author’s response: Thank you for your suggestion. We now included these data at Tables 2 and 3 (pages 5-7).
Reviewer 3 Report
Abstract: The abstract does not provide information on how the data analysis was carried out and provides little information on the results and conclusions. Please rewrite.
Introduction:
• On line 49, include a space between previous and 12.
• Indicate the source of figure 1.
methods:
• It would be easier to read if the contents were structured in different headings.
• Indicate on which date the data collection was carried out.
• what were the inclusion and exclusion criteria?
• How was the calculation of the sample size?
• There is a lack of information on how the data analysis was carried out. Please review it.
Results:
• Include the p-values in table 1
Discussion:
• The discussion is very limited. They should include more information.
Limitations:
• It is indicated that one of the strengths of this study is its longitudinal nature that allows establishing temporary associations. Although this is true, I consider that the sample size is not large enough to be able to make these statements.
Author Response
Abstract: The abstract does not provide information on how the data analysis was carried out and provides little information on the results and conclusions. Please rewrite.
Author’s response: Thank you for your suggestion. The abstract contains the following information about data analysis: “Mediation analysis was applied to verify whether the effect of the predictor variables (burnout and stress) on the dependent variable (number of pain sites in the last 12 months and 7 days) is due to an intervening variable (sleep quality)”.
Introduction:
- On line 49, include a space between previous and 12.
Author’s response: This phrase was rewritten.
- Indicate the source of figure 1.
Author’s response: The source was included at line 101 (page 3).
Methods:
- It would be easier to read if the contents were structured in different headings.
Author’s response: The headings were included (page 3-4).
- Indicate on which date the data collection was carried out.
Author’s response: The dates were included (line 106).
- what were the inclusion and exclusion criteria?
Author’s response: The inclusion and exclusions were included (lines 119-122).
- How was the calculation of the sample size?
Author’s response: The sampe size was not calculated, as the survey was open for paticipation for all interested workers. This information was included as a study limitation (lines 277-278).
- There is a lack of information on how the data analysis was carried out. Please review it.
Author’s response: Than you for your comment. Data analysis section was included at page 4 (lines 155-175).
Results:
- Include the p-values in table 1
Author’s response: We appreciate your suggestions. However, as Table 1 deals with sociodemographic and personal data description we do not have any p-value to report.
Discussion:
- The discussion is very limited. They should include more information.
Author’s response: We appreciate your suggestions. The Discussion was reviewed, including new references (pages 10-11).
Limitations:
- It is indicated that one of the strengths of this study is its longitudinal nature that allows establishing temporary associations. Although this is true, I consider that the sample size is not large enough to be able to make these statements.
Author’s response: We agree with you. Now we excluded this phrase, considering that the sample size is an important study limitation (lines 274-279).
Reviewer 4 Report
Dear authors,
I have some suggestions that must be addressed to improve your manuscript:
- in the abstract, you should mention the statistics methods used to obtain the conclusions
- the schemes in figure 1 are very simple and not easy to understand. Is it really necessary to use such visual information?
- in the introduction you should provide some background regarding MSDs, stress and burnout. Please find some information here and include the references on your manuscript:
* Gabriel, A. T., Madaleno, S., Kanazawa, F. & Ollay, C., 2023, Occupational and Environmental Safety and Health IV. Arezes, P. M., Baptista, J. S., Melo, R. B., Branco, J. C., Carneiro, P., Colim, A., Costa, N., Costa, S., Duarte, J., Guedes, J. C. & Perestrelo, G. (eds.). Cham: Springer, p. 259-268 10 p. (Studies in Systems, Decision and Control; vol. 449).
* Pimparel, A. J. C. S. M., Madaleno, S. Q., Ollay, C. & Gabriel, A. T., 2022, Occupational and Environmental Safety and Health III. Arezes, P. M., Baptista, J. S., Carneiro, P., Branco, J. C., Costa, N., Duarte, J., Guedes, J. C., Melo, R. B., Miguel, A. S. & Perestrelo, G. (eds.). Cham: Springer, p. 399-409 11 p. (Studies in Systems, Decision and Control; vol. 406).
* European Agency for Safety and Health at Work. (2022). Psychosocial risks and stress at work.
* O. K. Ornek and M. N. Esin, "Effects of a work-related stress model based mental health promotion program on job stress, stress reactions and coping profiles of women workers: a control groups study," BMC Public Health, vol. 20, no. 1, p. 1658, Nov. 2020, doi: 10.1186/s12889-020-09769-0.
* European Agency for Safety and Health at Work. (2022). Factsheet 22 - Work-related stress.
* G. La Torre et al., "Assessment of Anxiety, Depression, Work-Related Stress, and Burnout in Health Care Workers (HCWs) Affected by COVID-19: Results of a Case–Control Study in Italy," J. Clin. Med., vol. 11, no. 15, 2022, doi: 10.3390/jcm11154434.
- 1st paragraph of Results sections does not provide an adequate demographic characterization of the sample. Please revise the information.
- the first paragraphs of Discussion section are not adequate for this section. They are literature facts and should be reported in the Introduction section.
- the discussion of your results is very weak. I do not understand how you found most of your results and what is its contribution for your scientific conclusions. You should develop this section a lot. Also, I am wondering if some demographic characteristics (e.g: age, gender, working hours, type of healthcare professional) can influence the results. Also, I am curious if the quarterly can influence the results. Did you performed statistical tests to study this type of question? It would be necessary to verify if the data follow a normal distribution and then, based on the normality results, apply parametric/non parametric tests to understand the effect of some variables.
- the limitations text is under developed.
- the conclusions section is very poor. You should provide 2 or 3 more paragraphs. At least one to resume the importance of the topic and another to underline the scientific contribution of your work.
Author Response
Dear authors,
I have some suggestions that must be addressed to improve your manuscript:
- in the abstract, you should mention the statistics methods used to obtain the conclusions
Author’s response: Thank you for your suggestion. The abstract contains the following information about data analysis: “Mediation analysis was applied to verify whether the effect of the predictor variables (burnout and stress) on the dependent variable (number of pain sites in the last 12 months and 7 days) is due to an intervening variable (sleep quality)”.
- the schemes in figure 1 are very simple and not easy to understand. Is it really necessary to use such visual information?
Author’s response: We appreciate your suggestion. Figure 1 shows how mediation analysis was performed, according to the Hayes model, making the explanation of mediation easy to visualize and understand. Thus, we optioned to maintain it in this reviewed version of the manuscript.
- in the introduction you should provide some background regarding MSDs, stress and burnout. Please find some information here and include the references on your manuscript:
* Gabriel, A. T., Madaleno, S., Kanazawa, F. & Ollay, C., 2023, Occupational and Environmental Safety and Health IV. Arezes, P. M., Baptista, J. S., Melo, R. B., Branco, J. C., Carneiro, P., Colim, A., Costa, N., Costa, S., Duarte, J., Guedes, J. C. & Perestrelo, G. (eds.). Cham: Springer, p. 259-268 10 p. (Studies in Systems, Decision and Control; vol. 449).
* Pimparel, A. J. C. S. M., Madaleno, S. Q., Ollay, C. & Gabriel, A. T., 2022, Occupational and Environmental Safety and Health III. Arezes, P. M., Baptista, J. S., Carneiro, P., Branco, J. C., Costa, N., Duarte, J., Guedes, J. C., Melo, R. B., Miguel, A. S. & Perestrelo, G. (eds.). Cham: Springer, p. 399-409 11 p. (Studies in Systems, Decision and Control; vol. 406).
* European Agency for Safety and Health at Work. (2022). Psychosocial risks and stress at work.
* O. K. Ornek and M. N. Esin, "Effects of a work-related stress model based mental health promotion program on job stress, stress reactions and coping profiles of women workers: a control groups study," BMC Public Health, vol. 20, no. 1, p. 1658, Nov. 2020, doi: 10.1186/s12889-020-09769-0.
* European Agency for Safety and Health at Work. (2022). Factsheet 22 - Work-related stress.
* G. La Torre et al., "Assessment of Anxiety, Depression, Work-Related Stress, and Burnout in Health Care Workers (HCWs) Affected by COVID-19: Results of a Case–Control Study in Italy," J. Clin. Med., vol. 11, no. 15, 2022, doi: 10.3390/jcm11154434.
Author’s response: We appreciate your suggestions. We worked on the Introduction section to make it more fluid, clean and clear, facilitating understanding. Additional studies were included to improve our text (pages 1-2).
- 1st paragraph of Results sections does not provide an adequate demographic characterization of the sample. Please revise the information.
Author’s response: We appreciate your suggestion. We revised the data description at page 4 (lines 179-182).
- the first paragraphs of Discussion section are not adequate for this section. They are literature facts and should be reported in the Introduction section.
Author’s response: Thank you for your suggestion. We revised the Discussion section to make it clearer (pages 10-11).
- the discussion of your results is very weak. I do not understand how you found most of your results and what is its contribution for your scientific conclusions. You should develop this section a lot. Also, I am wondering if some demographic characteristics (e.g: age, gender, working hours, type of healthcare professional) can influence the results. Also, I am curious if the quarterly can influence the results. Did you performed statistical tests to study this type of question? It would be necessary to verify if the data follow a normal distribution and then, based on the normality results, apply parametric/non parametric tests to understand the effect of some variables.
Author’s response: Thank you for your suggestions. We included a new data analysis considering the 7 days symptoms. However, we understand that to include another data analysis could make the results even more complex and ouside the scope of our study.
- the limitations text is under developed.
Author’s response:Thank you for youe suggestion. The study limitations were reviewed (lines 274-279).
- the conclusions section is very poor. You should provide 2 or 3 more paragraphs. At least one to resume the importance of the topic and another to underline the scientific contribution of your work.
Author’s response:Thank you for youe suggestion. The study conclusion were reviewed (lines 282-287).
Reviewer 5 Report
Thank you for submitting the manuscript. I have read your manuscript with great interest and attention. The issue of burnout in healthcare workers is very important, as it is an emerging issue that involves many workers and which also has repercussions on patients. Your manuscript is well structured and valuable. However, I am convinced that it needs some revisions to make it more usable.The correlation between burnout and psychological and physical symptoms is an important issue to be investigated, as it not only gives us the dimension of the problem in terms of social costs, but directs us towards prevention and treatment.For this reason I would like you to treat this topic more extensively both in the introduction and in the discussion. In this regard, I suggest the following references that I would like you to use : doi: 10.3390/ijerph20032693. doi: 10.3390/healthcare10081370. doi: 10.13075/ijomeh.1896.01302.
I am convinced that a deeper reflection on the associated symptoms can help to better explain the rationale of the work.I hope my comments are helpful to you.
Kind Regards
Author Response
Thank you for submitting the manuscript. I have read your manuscript with great interest and attention. The issue of burnout in healthcare workers is very important, as it is an emerging issue that involves many workers and which also has repercussions on patients. Your manuscript is well structured and valuable. However, I am convinced that it needs some revisions to make it more usable. The correlation between burnout and psychological and physical symptoms is an important issue to be investigated, as it not only gives us the dimension of the problem in terms of social costs, but directs us towards prevention and treatment. For this reason I would like you to treat this topic more extensively both in the introduction and in the discussion. In this regard, I suggest the following references that I would like you to use : doi: 10.3390/ijerph20032693. doi: 10.3390/healthcare10081370. doi: 10.13075/ijomeh.1896.01302.
Author’s response: We appreciate your comments and suggestions. The suggested references were of great importance for the study.
I am convinced that a deeper reflection on the associated symptoms can help to better explain the rationale of the work.I hope my comments are helpful to you.
Author’s response: We appreciate your contributions. We hope that this reviewed version could better explain the rationale of the study.
Round 2
Reviewer 3 Report
Thank you very much for allowing me to revise the manuscript. It has improved substantially, but there are still certain aspects that need to be improved.
Methodology:
• The calculation of the sample size must be carried out. Although they have been included in the limitations, it must be ensured that the number of participants is sufficient to ensure adequate statistical power.
• Please, it would be convenient to carry out the hypothesis contrast tests and provide at least the p-values, confidence intervals or size of the effect. Although they are descriptive tables, these are data that must be included.
• It would be interesting to include the different professions of the participants. It could be included in the descriptive table (Table 1).
Author Response
Methodology:
- The calculation of the sample size must be carried out. Although they have been included in the limitations, it must be ensured that the number of participants is sufficient to ensure adequate statistical power.
Authors’ response: Thank you for your suggestion. We now included the sample size calculation at page 3 (lines 177-182).
- Please, it would be convenient to carry out the hypothesis contrast tests and provide at least the p-values, confidence intervals or size of the effect. Although they are descriptive tables, these are data that must be included.
Authors’ response: Thank you for your suggestion. We now included the results for the inferential tests at Tables 1 and 2.
- It would be interesting to include the different professions of the participants. It could be included in the descriptive table (Table 1).
Authors’ response: Thank you for your suggestion. We now included the professions at Table 1.